# Understanding the Molecular Regulatory Networks of Seed Size in Soybean

**DOI:** 10.3390/ijms25031441

**Published:** 2024-01-24

**Authors:** Ye Zhang, Javaid Akhter Bhat, Yaohua Zhang, Suxin Yang

**Affiliations:** 1Key Laboratory of Soybean Molecular Design Breeding, State Key Laboratory of Black Soils Conservation and Utilization, Northeast Institute of Geography and Agroecology, Chinese Academy of Sciences, Changchun 130102, China; zhangye@iga.ac.cn (Y.Z.); zhangyaohua@iga.ac.cn (Y.Z.); 2College of Advanced Agricultural Sciences, University of Chinese Academy of Sciences, Beijing 101408, China; 3Zhejiang Laboratory, Hangzhou 311100, China; javid.akhter69@gmail.com

**Keywords:** soybean, seed size, regulatory mechanism, genes, yield

## Abstract

Soybean being a major cash crop provides half of the vegetable oil and a quarter of the plant proteins to the global population. Seed size traits are the most important agronomic traits determining the soybean yield. These are complex traits governed by polygenes with low heritability as well as are highly influenced by the environment as well as by genotype x environment interactions. Although, extensive efforts have been made to unravel the genetic basis and molecular mechanism of seed size in soybean. But most of these efforts were majorly limited to QTL identification, and only a few genes for seed size were isolated and their molecular mechanism was elucidated. Hence, elucidating the detailed molecular regulatory networks controlling seed size in soybeans has been an important area of research in soybeans from the past decades. This paper describes the current progress of genetic architecture, molecular mechanisms, and regulatory networks for seed sizes of soybeans. Additionally, the main problems and bottlenecks/challenges soybean researchers currently face in seed size research are also discussed. This review summarizes the comprehensive and systematic information to the soybean researchers regarding the molecular understanding of seed size in soybeans and will help future research work on seed size in soybeans.

## 1. Introduction

Soybean [*Glycine max* (Linn.) Merr.] is an important oilseed crop that supplies half of the world’s vegetable oil and a quarter of the plant protein to the world population [1]. Throughout the history of crop breeding, substantial yield increases have been achieved for major grain crops such as rice, wheat, and maize; however, the yield gains of other important crops such as soybean have remained relatively stable or experienced slow growth. For example, in the highest soybean-consuming countries such as China, the past five decades have seen that soybean yield improvement efforts have been almost stagnant [2]. Hence, there is a great need for these countries as well as the world to increase domestic production to make self-sufficiency in soybean production. Breeders target different yield-related traits to increase soybean production. In this regard, seed size traits are an important trait that is directly related to the yield of soybeans [3]. Nitrogen fixation is vital for plant growth and yield, especially in legumes (such as soybeans) with their unique biological nitrogen-fixing ability [4]. Current research indicates that soybeans, through symbiosis with rhizobia, convert atmospheric nitrogen into ammonium [5], which crucially influences seed size and crop yield [6,7]. Optimizing this symbiotic advantage presents a promising avenue for agricultural yield improvement. Overall, the above results highlight the urgent need for the efficient and effective enhancement of soybean productivity [8].

Seeds are the most crucial organs for plant reproduction, and in conventional crop breeding the seed size has ranked among the top-priority agronomic traits. Over thousands of years of artificial selection and breeding, the seed size has been selected gradually, resulting in significantly different seeds of cultivated crops compared to their wild ancestors, thus indicating parallel selection processes [9]. Unraveling the regulatory mechanisms underlying seed size is not only a central question in plant embryonic development but it also holds significant importance for understanding crop adaptability and yield [10]. As the global population continues to surge which is accompanied by scarce land and climate change, the demand for improved crop cultivars with higher seed size and crop yields is poised to significantly escalate [11].

The seed size of soybeans are relatively complex trait governed by polygenes with minor effects on the traits as well as are influenced by the environment and genotype x environment interactions besides possessing low heritability [3]. Although hundreds of quantitative trait loci (QTL) associated with seed size and shape have been identified and are being documented in the SoyBase https://www.soybase.org/ (accessed on 20 January 2024); however, the true functional genes underlying these QTLs have been rarely identified and are very scarce to be applied in practical soybean breeding [12]. Identification of functional genes associated with the seed size traits in soybeans would significantly enhance the efficiency of soybean molecular breeding efforts. Moreover, current research on the molecular regulatory networks governing seed size in plants has predominantly focused on rice and *Arabidopsis thaliana* (*A. thaliana*), with limited studies specifically conducted on soybeans [13]. Seed size research on rice has made substantial progress and has used the functional genes related to seed size in practical breeding to harness the true potential of crop breeding [14]. However, due to significant differences between rice (being a monocot) and soybean (being a dicot), their seed characteristics differ considerably. Moreover, previous studies based on seed size in *A. thaliana* have mostly remained in the experimental analysis stage and are still far from practical application in crop production. Hence, the identification of key genes controlling seed size and unravelling their molecular regulatory networks is crucial for boosting soybean yield. The recent progress made in genomics, phenomics and other omics approaches has provided exciting opportunities to conduct precision breeding at higher efficiency [15].

The existing research in this field is highly scattered in the current literature, necessitating a comprehensive review to provide a clear and systematic framework for researchers and readers. Therefore, this article summarizes the recent research progress made in the molecular regulatory mechanisms of seed size, as well as molecular pathways involved in seed size regulation in soybeans, and the major challenges and prospects in this field. It also presents, for the first time, a systematic compilation of confirmed or finely mapped genes and their accession numbers related to soybean seed size regulation, aiming to provide the basis for their future use in practical soybean breeding leading to improved soybean cultivars with improved seed size and yield.

## 2. The Seed Size Diversity of Soybean Makes It an Ideal Model Plant for Study

Besides the importance of genetic diversity for trait improvement in plant breeding, it also has great significance for the study of genetic evolution as well as the molecular regulatory mechanism. Hence, species possessing larger genetic diversity are serving as the model species in the study of crop evolution and genetic mechanisms. Among the flowering plants, leguminous plants constitute one of the largest and most diverse plant families, with approximately 20,000 species. The Fabaceae family is the largest subfamily within legumes and it encompasses a majority of model species such as soybean [16]. The soybean plants possess a unique advantage in the study of seed morphology. They exhibit a diverse range of seed sizes [17], and their embryos display diverse morphological forms (Figure 1).

As early as around 5000 BC mankind started to cultivate soybeans, and conserved as well as managed the rich source of soybean germplasm resources. The abundance of genetic diversity allows soybean breeders and researchers to conduct genetic analysis as well as functional studies using different soybean varieties and mutants. By using recent genetic and omics techniques combined with artificial intelligence (AI) based models such as machine learning (ML) and deep learning (DL) the soybean researchers will be able to unravel regulatory molecular mechanisms that influence seed size. Moreover, soybean exhibits extensive genetic variation in traits such as seed size (Figure 1). The seed size differences among different soybean varieties can exceed fivefold, hence it makes the soybean an ideal model plant for studying seed size.

## 3. Molecular Regulatory Network Underlying Seed Size in Soybean

The molecular regulatory network controlling seed size in soybeans is a complex system involving interactions at multiple levels, such as gene regulation, hormone regulation, nutrient regulation, and signal transduction pathways (Table 1). These regulatory factors collectively modulate the development of the embryo and endosperm, ultimately influencing seed size traits. Furthermore, previous studies on seed size regulatory mechanisms have mostly focused on internal factors, neglecting the profound impact of environmental factors on soybean seed development. For instance, soybean is highly sensitive to photoperiod, and the duration as well as intensity of light exposure can affect the process of seed development. In recent years, the influence of environmental factors on soybean seed development has gained widespread attention among researchers [18,19].
ijms-25-01441-t001_Table 1Table 1Genes associated with the development of soybean seed size/weight.PositionProtein NameProtein CategoryAccession NumberPositive/NegativeMolecular MechanismReferenceTranscriptional regulatorsBBMAP2 family transcription factors*Glyma.09G248200*positivePositively regulates the process of seed embryogenesis and development and plays a decisive role in seed maturationOuakfaoui, 2010 [20]GmAP2-1AP2 family transcription factors*Glyma.01G188400*positiveAll three together regulate seed size and grain weight by affecting seed length, width, and areaJiang, 2020 [21]GmAP2-4AP2 family transcription factors*Glyma.13G329700*positiveJiang, 2020 [21]GmAP2-6AP2 family transcription factors*Glyma.08G279000*positiveJiang, 2020 [21]PP2C-1Phosphatase 2C-1*Glyma.17G221100*positiveIncreases bead cell size and activates seed development-related genes; interacts with *GmBZR1*Lu, 2017 [22]GmBS1Tify domain*Glyma.10G244400*negativeNegative regulation of primary cell proliferation by suppressing the expression of *GIF1* and *GRF5*Ge,2016 [23]GmBS2*Glyma.20G150000*negativeGmCYP78A5cytochrome P450 family protein CYP78A10*Glyma.05G00220*positivePrediction by RNA-seq data analysis may correlate with seed sizeDu, 2017 [24]CYP78A10Cytochrome P450*Glyma.05G019200*positivePredicted by evolutionary correlation analysis to be extremely correlated with seed size, but leads to lower pod numbersWang, 2015 [25]CYP78A72Cytochrome P450*Glyma.19G240800*positiveOverexpression increases seed size and is functionally redundant with two other homologous genesZhao, 2016 [26]GmWRKY15aWRKY family transcription factor*Glyma.05G096500*positiveDifferential expression in soybean pods is significantly associated with CT repeat variation during soybean domesticationGu, 2017 [27]Phytohormone signalling and homeostasis GmBZR1transcription factor of the BR signalling pathway*Glyma.17G248900*positiveConserved regulation by BR signalling promotes phosphorylation/dephosphorylation ratiosZhang, 2016 [28]GmGA20OXgibberellin biosynthetic enzymes*Glyma.07G081700*positiveHighly selected during domestication to increase the rate of GA biosynthesisLu, 2016 [29]GmJAZ3protein TIFY 6a-related*Glyma.09G123600*positivePromotes seed size/weight and other organ sizes in stable transgenic soybean plants by increasing cell proliferationHu, 2023 [30]SGF14fprotein SGF14f *Glyma.02G115900*unknownMay regulate the balance of GA and ABA signalling and determine embryogenesis and seed germinationSchooheim, 2009 [31]GmFAD3microsomal omega-3 fatty acid desaturase*Glyma.18G062000*negativeAssociated with fatty acid synthesis, which inhibits jasmonic acid accumulation, and silencing leads to greater seedSingh, 2020 [32]unnamedethylene-response factor C3*Glyma.19G163900*positiveAHP can promote cytokinin signaling with multiple functionally redundant genesClaire, 2019 [33]unnamedHpt domain*Glyma.19G151900*positivePossible involvement in cytokinin-mediated seed size and weight regulatory networksAssefa1, 2019 [34]GmPSKγ1PSK-like peptide*Glyma.02G126200*positiveNovel peptide hormone whose tyrosine is sulfated induces embryonic cell expansion for seed growth Yu, 2019 [35]Metabolic pathwayGmST01UDP-galactose4-epimerase; domain 1*Glyma.08G109100*positiveRegulation of cell division and amplification patterns in soybean to determine seed shapeLi, 2022 [36]GmST05Phosphatidylethanolamine-binding Protein*Glyma.05G244100*positiveregulates seed size and affects oil and protein content, which may affect the transcription of GmSWEET10aDuan, 2022 [37]GmSWEET10asugar efflux transporter SWEET39*Glyma.15G049200*positiveThe former is strongly selected in domestication and the latter is being selected, both contribute to the sugar distribution from the seed coat to the embryo by transporting sucrose and hexose, thus increasing oil content and seed size Wang, 2020 [38]GmSWEET10bsugar efflux transporter SWEET24*Glyma.08G183500*positiveCIF1Cell wall invertase inhibitors*Glyma.17G036300*negativeCoordination of seed maturation by post-translational fine-tuning of sucrose metabolism and library strength by CWITang, 2017 [39]*GmDREBL*Caskin/Ankyrin repeat-containing protein*Glyma.12G11150*positiveInvolved in the regulation of fatty acid accumulation by controlling the expression of WRI1 and its downstream genesZhang, 2016 [40]*GmNSS*PEPTIDASE-C1 DOMAIN-CONTAINING PROTEIN*Glyma.08G309000*positiveRegulation of the size of the outer bead cover cellZhang, 2023 [41]Other regulatorsGmDof4DOF ZINC FINGER PROTEIN DOF1.1-RELATED*Glyma.17G081800*positiveIncreasing lipid content in soybean seeds by upregulating genes related to fatty acid biosynthesis and direct binding downregulation of the storage protein gene CRA1 by the cis-DNA element in its promoter regionWang, 2007 [42]GmDof11DOF ZINC FINGER PROTEIN DOF1.1-RELATED*Glyma.13G329000*positiveunnamedRING FINGER AND CHY ZINC FINGER DOMAIN-CONTAINING PROTEIN 1*Glyma.17G202700*unknownPrediction by GWAS analysis may correlate with seed sizeAssefa1, 2019 [34]GmSSS1tetratricopeptide repeat protein, TPR*Glyma.19G196000*positiveExerts a positive impact on cell expansion and cell division, thus regulating the ultimate size of soybean seedsZhu, 2022 [43]


### 3.1. The Ubiquitin-Proteasome Pathway

Protein ubiquitination controls many aspects of cellular processes by affecting protein stability, activity, and localization [44]. The ubiquitination process involves a series of specialized enzymes such as ubiquitin-activating enzymes (E1s), ubiquitin-conjugating enzymes (E2s), and ubiquitin ligases (E3s) [45]. Ubiquitin or ubiquitin chains can be removed by deubiquitinating enzymes. Recent studies have revealed the important role of the ubiquitin-proteasome pathway in the regulation of seed size. For example, *DA1* is a ubiquitin receptor in *A. thaliana* that contains two ubiquitin-interacting motif (UIM) domains viz., a LIM domain and a C-terminal peptidase domain [46]. *DA1* negatively regulates seed and organ growth by modulating cell proliferation in the ovule integument [47]. Additionally, *DA1* can interact with its closest homolog, *DAR*, to control seed size.

In soybean, a gene encoding an E3 ligase zinc finger protein (*Glyma.17G202700*) is highly expressed in the seed coat of large seeds. Additionally, a gene encoding an E2 conjugating enzyme phosphatase 2 (*Glyma.07G196500*) is expressed at seven-fold higher levels in the seed coat of large seeds compared to small seeds. These findings suggest the key role of the ubiquitin-mediated protein degradation pathway in the regulation of seed size in soybean [24].

### 3.2. Plant Hormone Signaling Pathways

Plant hormones have been documented to play a crucial role in the regulation of seed development. Different hormones exert their specific effects on various aspects of seed development, including seed size, embryo development, endosperm development, and seed dormancy.

#### 3.2.1. Brassinosteroids

Brassinosteroids (BRs) are a class of plant hormones that play an important role in various aspects of plant growth and development [48]. They are involved in regulating cell elongation, cell division, differentiation, and senescence [48]. The BRs have been demonstrated to primarily regulate seed growth through maternal tissues [49]. Additionally, BRs can influence endosperm development and govern the final size of seeds. For example, in the case of rice and *A. thaliana*, non-functional mutations in *BR* biosynthesis genes lead to the formation of shorter and smaller seeds, this suggests an important role of BRs in the regulation of seed growth [50]. It has been revealed that the small-grain phenotype of *BR*-deficient mutants is primarily caused by the reduced cell size in the lemma/palea, suggesting that BR regulates grain size by promoting cell expansion in the glumes [51]. Genetic analysis in *A. thaliana* has revealed that BR also controls seed size and shape by affecting endosperm development [52].

In soybeans, it has been reported that the BRI1-associated receptor kinase is highly expressed in the seed coat, and has been identified as a key gene in early seed coat maturation [24]. Lu et al. [22] reported that the *PP2C-1* gene regulates the BR signalling pathway in soybeans and controls the seed size [22]. They demonstrated that *PP2C-1* mainly promotes the expansion of epidermal cells to enhance seed growth in soybeans. This gene interacts with the core transcription factor *GmBZR1* in the BR pathway, leading to its de-phosphorylation and activation of downstream transcription factors. By regulating the expression of related genes, it ultimately increases seed size and weight. The *PP2C-1* gene is primarily found in wild soybeans, and around 40% of cultivated soybeans lack this gene. Therefore, introducing this gene into cultivated soybeans can be expected to improve soybean seed size and yield considerably.

Although BR and many components of the BR signalling pathway have been implicated in seed size control, the mechanisms of BR-mediated promotion of seed growth have remained unexplored. Firstly, our understanding of the genetic interactions between the identified regulators of seed size and BR-related factors is still limited, resulting in a fragmented understanding of BR’s role in seed size control. Secondly, some BR signalling molecules can influence seed size by controlling the number of cell divisions in the maternal tissues, while others affect cell expansion, suggesting that BR might regulate seed growth through different downstream pathways. Furthermore, evidence from rice suggests that BRs regulate grain size by influencing the size of the hull [51], whereas, in *A. thaliana*, BRs affect seed size by modulating endosperm development. This indicates that the mechanism of BR-mediated regulation of seed size varies among different plant species. Hence, substantial research efforts are needed to elucidate the detailed regulatory mechanism of BR in governing seed size in soybeans [53].

#### 3.2.2. Gibberellins

Gibberellins (GAs) are the crucial hormones playing key roles throughout the entire life cycle of plants including from embryonic development to maturation of individuals. Swain (1995) reported that a decrease of GA levels in the endosperm of pea (*Pisum sativum*) leads to reduced seed weight as well as increased production of sterile seeds [54]. Moreover, the application of GA biosynthesis inhibitors to developing pea seeds resulted significant reduction of seed weight, although mature seeds are still formed. Additionally, a core enzyme involved in GA synthesis, *GA20ox*, also influences the seed size [29]. Overexpression of *GA20ox* has been revealed to significantly increase seed size in transgenic soybean plants [22]. It has been documented that the expression level of this gene is significantly higher in cultivated soybeans compared to wild soybeans, and it is closely linked to previously identified major seed weight quantitative trait loci (QTL) [22]. Studies have revealed that the increase in *GA20ox* expression in cultivated soybeans is associated with variations in the promoter region of the gene [29]. Specifically, cultivated soybean possesses specific haplotypes in the promoter region of the *GA20ox* gene, characterized by variations in the long terminal repeat (LTR) sequences located at the long end of the promoter region. These variations in the promoter region of the *GA20ox* gene affect seed size in cultivated soybeans.

#### 3.2.3. Cytokinin

Cytokinins have an important role in the regulation of cell division of endosperm cells during seed development; hence, they control the seed size by influencing endosperm development. Moreover, the level of cytokinins is increased during embryogenesis in the seeds of many plant species, including legumes. For example, Bandyopadhyay et al. [49] reported that the common cytokinin *viz.,* isopentenyladenine (iPR) is associated with cell proliferation during alfalfa seed development [55]. In soybeans, Assefa et al. [50] identified a candidate gene (*Glyma.19G151900*) that encodes a histidine phosphotransfer protein, and this gene regulates seed weight in soybeans [34]. Previous studies in *A. thaliana* have documented that histidine phosphotransfer proteins (AHPs) transfer the phosphorylation signal from cytokinin receptors *viz., A. thaliana* histidine kinases (AHKs) to *A. thaliana* response regulators (ARRs), thereby mediating cytokinin responses in cells [56]. Mutations in *AHP A. thaliana* genes result in larger seeds, suggesting that the soybean homologous gene *Glyma.19G151900* might be also involved in the cytokinin-mediated regulation of soybean seed weight [33]. Hu et al. [30] reported that *GmJAZ3* (*Glycine max JASMONATE-ZIM DOMAIN 3*) promotes increased cell proliferation and enhanced seed size/weight as well as organ size in stable transgenic soybean plants [30]. The *GmJAZ3* interacts with *GmRR18a* and *GmMYC2a* to inhibit their transcriptional activation ability on the cytokinin oxidase gene *GmCKXs*, leading to increased protein content and decreased fatty acid content in soybean seed components, ultimately promoting soybean seed size and weight.

Although some understanding regarding the involvement of plant hormone signalling in seed size regulation has been achieved, the precise mechanisms by which these hormones regulate seed development remain unexplored. The molecular mechanisms underlying plant hormone-mediated seed development, particularly the regulatory mechanisms that maintain homeostasis during seed development need to be further investigated. Exploring key genes involved in hormone metabolism, transport, signal transduction, or homeostasis in leguminous plants and studying their roles in seed development will be crucial for unravelling the mechanisms underlying soybean seed size traits.

#### 3.2.4. Phytosulfokine

Phytosulfokine (PSK) is a novel peptide growth regulator and has been considered a new peptide hormone that regulates various aspects of plant growth and development [57]. The precursor of PSK-α is encoded by a small gene family which is widely distributed across the plant kingdom. Studies have revealed that overexpression of the PSK-α precursor gene significantly enhances cell proliferation and promotes root, hypocotyl, and leaf growth in *A. thaliana* [58]. Yu et al. [35] discovered a novel PSK-encoding gene, *GmPSKγ1*, in soybean, which encodes a PSK-α analogue called PSK-γ [35]. *GmPSKγ1* is mainly expressed in developing seeds of soybeans and increases seed size in transgenic *A. thaliana*. Furthermore, it was revealed that the expression of several genes related to cell expansion was altered in transgenic *A. thaliana*, including the upregulation of two genes, *Cel2* and *SAUR74*, which promote cell expansion, and the downregulation of two peroxidase family genes *viz., POD-49* and *POD-64*, which affect cell wall stability. Therefore, PSK-γ might be involved in promoting embryonic cell expansion growth by regulating cell wall stability [35].

### 3.3. Regulation of Seed Size by Transcription Factors

Transcriptional regulation is crucial for plant growth and development, and several transcription factors have been identified as the key regulators of seed size. These transcription factors, along with transcriptional co-activators and regulators involved in chromatin modification, play important roles in determining seed size in plants.

In soybeans, several transcription factors have been reported to be involved in seed size regulation. This includes important gene families such as *WRKY* and *CP450* [24]. These transcription factors likely control seed size by regulating the expression of genes that are involved in cell proliferation, cell expansion, and other processes during seed development. Their precise mechanisms of action and specific target genes are still being investigated to gain a better understanding of their roles in the regulation of seed size in soybeans.

#### 3.3.1. APETALA2 Transcription Factor

The *AP2*/*ERF* gene family is one of the largest transcription factor families in plants. It generally contains one or two conserved AP2 domains and includes three subfamilies: ERF, AP2 and RAV [59]. The AP2 subfamily can be further divided into the AP2 and ANT groups [60]. AP2 transcription factors are well known for their crucial role in determining floral organ identity in *A. thaliana* [61]. However, AP2 family members are not only involved in regulating plant flower formation and development, synthesis of fatty acids in seeds, and stress resistance, but also regulate ovule and seed coat development, hence determining the seed size and weight [62]. The suppression or inhibition of AP2 function has been reported to increase the size of the outer integument cells, which indicates the role of AP2 in seed size regulation [63]. A specific AP2-type transcription factor, *SMALL ORGAN SIZE1* (*SMOS1*), has been identified as a gibberellin-dependent regulator of grain and organ size in rice [64]. The *smos1* mutant exhibits reduced organ and grain size due to decreased cell size and abnormal microtubule orientation. Recent studies have shown that *SMOS1* forms a complex with DLT that in turn regulates rice brassinosteroid (BR) response and grain size, thus suggesting the role of *SMOS1* in controlling rice grain size by integrating auxin and BR signalling [65,66].

In the case of leguminous plants, Confalonieri et al. [67] reported that the specific expression of *ANT* (*AINTEGUMENTA*) in the seeds of *Medicago truncatula* (*M. truncatula*) not only resulted in larger seeds but also enhanced seed germination rate. These authors revealed that a gene driven by the seed-specific promoter USP (unknown seed protein) in *M. truncatula* seeds leads to the expansion of the storage parenchyma cells in the cotyledon as well as a significant increase in the size of vacuoles, thus producing the large-seeded phenotype. The larger parenchyma cells in the cotyledons positively affect the extent and dynamics of water absorption in *M. truncatula* seed tissue, which in turn results in *M. truncatula* seeds overexpressing *ANT* having higher moisture content, which is beneficial for seed germination [67].

The AP2/ERF transcription factor family member *BABY BOOM* (*BBM*) is a key regulator of plant cell totipotency and plays multiple roles in plant cell proliferation and growth [68]. Recently, a homolog of *BBM* called *GmBBM1* has been identified in soybeans, and this gene has been reported to possess an important role in somatic embryogenesis and embryonic development [20]. Furthermore, a candidate gene, *Glyma.19G163900*, encoding an AP2-domain protein was reported to be involved in the regulation of seed size and was found on chromosome 19 of soybean. Jiang et al. [21] identified several seed development-related genes within the AP2/ERF family through genome-wide sequence analysis in soybeans. Among them, the genes *GmAP2-1*, *GmAP2-4*, and *GmAP2-6* were identified as important genes regulating soybean seed size development as well as showed a positive regulatory effect on seed weight and size [21].

#### 3.3.2. WRKY Transcription Factor

WRKY proteins are a unique class of transcription factors found in plants. They are named after the highly conserved WRKY domain they possess [69]. WRKY transcription factors have multiple biological functions and are involved in various processes of plant growth, development, metabolism, and responses to biotic and abiotic stresses. They can act as both transcriptional activators and repressors [70]. In *A. thaliana*, *SHB1* (*SHORT HYPOCOTYL UNDER BLUE1*) promotes endosperm proliferation, leading to enlarged seeds, and upregulates the expression of the WRKY transcription factor gene *MINI3* (*MINISEED3*) and the LRR receptor kinase gene *HAIKU2* (*IKU2*) [71]. *TRANSPARENT TESTA GLABRA2* (*TTG2*) is a key gene encoding a WRKY transcription factor in *A. thaliana* [72], and *ttg2* mutants were shown to exhibit reduced cell length in the seed coat that results in small and rounded seeds. Hybridization studies have revealed that *TTG2* regulates seed growth through maternal tissues [73]. In addition, the foxtail millet gene *LOOSE PANICLE1* (*LP1*) encodes a WRKY transcription factor, and the loss of *LP1* function leads to loose panicles and enlarged seeds, indicating that *LP1* negatively regulates seed size in foxtail millet [74].

In soybean expression level of *WRKY15* has been reported to show correlations with the variations in seed size in wild and cultivated soybeans, suggesting its role in the regulation of seed size [27]. Gu et al. [27] reported the role of the *SoyWRKY15a* gene in the regulation of seed size in soybeans [27]. The coding sequence region of this gene is nearly identical in cultivated and wild soybeans, but its expression level is significantly higher in cultivated soybeans. By comparing the *WRKY15a* gene sequences between the wild and cultivated soybeans, it was revealed that variation in the copy number of CT microsatellite sequence in the non-coding region 5’UTR is closely associated with gene expression, and this variation likely arose during the domestication process of soybean. Therefore, this CT microsatellite locus can be used as a functional molecular marker in soybean breeding.

#### 3.3.3. Cytochrome P450 (CYP) Transcription Factor

The cytochrome P450 (CYP) gene family is a group of transcription factors found in various organisms, including plants. They are involved in multiple types of biological processes such as metabolism of endogenous compounds, detoxification of xenobiotics, and synthesis of secondary metabolites. In plants, the CYP gene family plays an important role in diverse physiological and developmental processes, including hormone signalling, and defence responses, and has been identified as the regulator of organ size and development. For example, overexpression of *CYP78A6* significantly increases seed size in *A. thaliana*, while *CYP78A6* mutant plants exhibit a smaller seed phenotype [75]. Subsequently, studies have revealed that *EOD3*/*CYP78A6* regulates seed size by promoting the formation of more and larger cells in the seed coat.

In soybean, RNA sequencing analysis by Du et al. [24] identified the enrichment of the mRNA of CYP family gene *KLU* (*Glyma.02G119600*) in the seed coat of large-seeded genotypes [24]. The closely related homolog, *Glyma.05G019200* (*GmCYP78A10*) has been observed to show a positive correlation with soybean seed size and pod number. Furthermore, overexpression of *Glyma.05G019200* increased soybean seed size, thus confirming its role in the regulation of seed size in soybeans.

#### 3.3.4. TIFY Transcription Factor

The TIFY protein family is a diverse and significant group of plant proteins playing key roles in multiple biological processes. These proteins are characterized by the presence of a conserved domain known as the TIFY domain, which is essential for their function [76]. TIFY proteins are involved in the regulation of plant growth, development, and responses to multiple environmental stimuli, including both biotic and abiotic stresses [77,78]. They participate in signalling pathways that mediate plant defence against pathogens, modulate hormone signalling, and regulate plant responses to diverse stress conditions.

Ge et al. [23] reported a negative regulatory gene called *BS1* (*BIG SEED1*) in *M. truncatula*, which controls organ size [23]. In soybeans, the orthologous genes *GmBS1* (*Glyma.10G244400*) and *GmBS2* (*Glyma.20G150000*) exhibit similar pathways in the regulation of seed size and weight [23]. Subsequent experimental evidence confirmed that *BS1* significantly suppresses the expression levels of *GIF1* and *GRF5* in developing organs, and these two factors are positive regulators of primary cell proliferation, this implies that the deletion of *BS1* can increase the seed size in soybeans [23].

*BS1* encodes a member of the II subgroup of the TIFY transcription factor family and lacks any known DNA-binding domains. It shares homology with two tandemly duplicated genes in *A. thaliana* viz., *At4g14713* (*PEAPOD1* or *PPD1*) and *At4g14720* (*PPD2*) [79]. These genes were revealed to regulate leaf and silique size as well as shape but not seed size. Additionally, certain TIFY proteins interact with Novel Interactors of JAZ (NINJA) [80], a bridging protein that interacts with transcriptional corepressor TOPLESS (TPL) and TOPLESS-related proteins (TPR) to suppress downstream gene expression [81]. *BS1* can form a repressor complex with NINJA, thereby negatively controlling the expression of downstream target genes.

### 3.4. Photoperiod Regulation of Soybean Seed Size

The photoperiod i.e., the duration of light and dark periods in a day, can have an impact on seed size in plants [82]. Different plant species showed varied responses to photoperiod, and it can influence different aspects of seed development and growth. Plants have developed effective strategies to adapt to environmental changes throughout their evolution. For instance, plants perceive variations in daylight duration, allowing them to undergo the transition to flowering during seasons most conducive to successful reproduction [83]. Flowering time is a critical trait in domestication, and increased knowledge of flowering regarding its genetic basis and molecular mechanism will greatly enhance crop plants’ adaptation to the environment [84].

*CONSTANS* (*CO*) acts as a central regulator in the photoperiodic flowering pathway, and *CO* orthologs have been identified in various plant species [85,86,87,88]. Yu et al. [84] discovered that under favourable conditions of reproductive growth, plants tend to suppress the transcription of *APETALA2* (*AP2*) through *CO*, thereby regulating the proliferation of seed coat epidermal cells and promoting larger seed production [89]. Under conditions favouring vegetative growth, CO protein becomes unstable, leading the plants to prioritize nutrient acquisition and generate smaller seeds by attenuating the inhibitory effect of *CO* on *AP2*. When *CO* is mutated, the seed size of the plant becomes unresponsive to photoperiod. Thus providing the first insight into the core regulatory module governing seed size in response to photoperiodic cues.

### 3.5. Other Regulatory Pathways of Soybean Seed Size

Nguyen et al. [90] reported and characterized the *GmKIX8-1*, a gene that encodes a nuclear protein regulating organ size in soybeans. *GmKIX8-1* encodes a conserved KIX domain protein that is homologous to *AtKIX8* in *A. thaliana*, which limits organ growth by modulating cell proliferation in meristematic tissues [90]. Previous studies have revealed that the *KIX*-*PPD*-*MYC*-*GIF1* module controls seed size in *A. thaliana* by inhibiting cell proliferation in the outer integument during the development of embryo sac and early seed [91], this regulatory mechanism may also exist during seed development in soybean.

Flavonoids viz., anthocyanins, proanthocyanidins, flavonols, and isoflavones, are the most important components affecting seed coat colour [92]. Zhang et al. [41] identified a novel gene, *Glyma.08G309000*, from the mutant *S006*, which was named *Novel Seed Size* (*NSS*) [41]. This gene is associated with brown and small-seeded phenotypes. This study indicated a significant increase in anthocyanin accumulation in the *S006* mutant leading to pigmentation in the seed coat. Moreover, the outer integument cell area was significantly reduced in the *S006* mutant, thus negatively impacting seed size in soybeans. This study provides evidence that the brown seed coat colour might be attributed to elevated expression of *chalcone synthase 7/8* genes, while decreased expression of *NSS* contributes to reduced seed size. These findings suggest that the *NSS* gene represents a novel regulator of seed development. The *NSS* gene encodes a protein of unknown function, but it contains a peptidase-c1 domain resembling a potential DNA helicase RuvA subunit, indicating a possible involvement in apoptosis.

The plant cell wall is a highly complex structure composed of structural proteins, enzymes, and various polysaccharides as well as pectin [93], which plays a crucial role in the primary cell wall of plants [94]. Li et al. [36] reported a semi-dominant locus named *ST1* (*Seed Thickness 1*) [36] and successfully identified the underlying major functional gene of *ST1* as *Glyma.08G109100*. This gene was documented to regulate soybean seed thickness and encodes a UDP-D-glucuronic acid 4-epimerase, which regulates the production of UDP-rhamnose and promotes pectin biosynthesis. Thus, it may determine seed shape by modulating cell division and expansion patterns in soybeans. Interestingly, this morphological variation simultaneously increases seed oil content. Together with the upregulation of sugar metabolism regulated by *ST1*, soybean has become a major oilseed crop, suggesting a strong selection for this gene during domestication. Additionally, Tang et al. [39] discovered that the soybean transformation inhibitor *GmCIF1* participates in controlling seed maturation by specifically inhibiting the activity of cell wall invertase (CWI) [39]. Silencing of *GmCIF1* expression elevates the CWI post-translational levels to finely tune sucrose metabolism and sink strength, thereby coordinating the process of seed maturation and increasing seed weight, as well as accumulating hexoses, starch, and proteins in mature seeds.

The 100-seed weight is considered one of the pivotal domestication traits that greatly influences soybean yield. Nevertheless, its elusive genetic foundation remains enigmatic. Zhu et al. [43] elucidated a *soybean seed size 1* (*sss1*) mutant with enlarged seeds in comparison to its wild-type counterpart [43]. These authors showed that candidate gene *GmSSS1* (*SOYBEAN SEED SIZE 1*) encodes a SPINDLY homolog that resides within a well-defined quantitative trait locus (QTL) hotspot on chromosome 19, which underwent intense selection during the cultivation of soybeans. Deletion of *GmSSS1* causes reduced seed size, while its overexpression results the larger seeds, subsequently augmenting the 100-seed weight of a soybean. Further investigations indicate that *GmSSS1* exerts a positive effect on cell expansion and cell division, thus regulating the ultimate size of soybean seeds (Figure 2).

## 4. Challenges and Future Prospects of Soybean Seed Size Research

Three important challenges hinder the improvement of seed size in soybeans. (i) Complex inheritance of the seed size traits; (ii) negative correlation among the seed size and seed shape traits with other yield-related traits such as number of seeds per pod and number of pods per plant; and (iii) complex regulatory molecular mechanism of seed size. Firstly, the seed size of soybean is a complex trait, that is governed by multiple genes distributed throughout the genome [3]. The seed size traits possess low heritability as well as are highly influenced by the environmental factors and the interactions between genotype and environment. [3]. These complexities in trait variation have improved seed size a difficult task. The marker-assisted selection (MAS) uses the marker-trait associations initially identified through QTL or association mapping approaches and was considered to be effective for the improvement of complex traits such as seed size [95]. However, in the past two decades, more than 400 QTLs have been identified for the seed size and seed shape traits in soybean https://www.soybase.org/ (accessed on 20 January 2024). Most of these QTLs are not validated and are not used in soybean improvement [3]. Moreover, traditional MAS uses only the major QTLs/genes in crop breeding, but most of the genetic variation of seed size traits is governed by minor genes, thus making this variation unavailable for trait improvement [2]. The more recent approach of genomics-assisted breeding (GAB) i.e., genomic selection (GS) incorporates all the markers distributed across the whole genome into the model to generate a prediction that was the total of all genetic effects, regardless of how many minor and major, and thus the substantial variation contributed by the minor effect loci are made available for breeding [96]. In the past, mostly the linear models were used in GS and these models are unable to capture the nonlinear interactions of complex traits; hence, this leads to considerable loss of this variation which contributes a significant proportion for complex traits and thereby impedes crop improvement [97]. In this context, the recent emergence of artificial intelligence (AI) techniques has opened windows for the use of non-linear models in plant breeding, and these models can capture the nonlinear and epistatic interactions in GS thus making this variation available for GAB [97]. Additionally, the approaches of AI technology such as machine learning (ML) and deep learning (DL) have allowed the use of complex, big and multiomics datasets simultaneously such as genomics, phenomics, single-cell genomics, enviromics and transcriptomics etc., (Figure 3) for cultivar selection in predictive breeding [97,98]. Although some challenges persist in the AI-based models, particularly the statistical and software, these challenges will be resolved soon.

Secondly, the negative correlation among the seed size, seed shape and number of seeds per pod as well as the number of pods per plant limits the overall yield improvement as well as seed size research in soybeans [3,25]. In soybeans, the number and size of seeds within each pod are variable. Wang et al. [25] have shown that there is a trade-off between the number of pods per plant and seed size in soybean, for example, they have revealed that overexpression of the *P450* gene viz., *GmCYP78A10* increases seed size but reduces pod number per plant, resulting in non-significant yield differences; however, the underlying mechanisms causing this reduction in pod number per plant are still not well understood. Moreover, a significant negative correlation has been reported to exist between seed shape and seed size traits [3]. The only way to break this negative correlation is to use the non-correlated genes/QTLs in genomics-assisted breeding (GAB) [99]. However, it needs a great effort first to identify the non-correlated genes/QTLs for the seed size, seed shape and number of pods per plant. In future research emphasis will be made on the identification of non-correlated genes/QTLs for the above traits, and their utilization in GAB will have a tremendous impact to improve simultaneously the seed size, seed shape and number of pods per plant as well the significant improvement of overall yield in soybean.

Lastly, the seed size is regulated by the complex network of regulatory pathways. As discussed above the multiple upstream and downstream pathways are involved in the regulation of the seed size in soybeans. However, there is a dearth of systematic analysis of the molecular pathways upstream and downstream of seed size regulation in soybeans. To date, only a few genes for seed size have been isolated and their molecular mechanisms have been elucidated [3]. Hence, there is an urgent need to place more emphasis on exploring the in-depth and detailed molecular regulatory mechanisms of seed size in soybeans. In this regard, the recent advances in genomic technologies and resources as well as gene editing techniques, offer new opportunities to unravel the molecular mechanism of seed size in soybeans. To this end, transcriptomic profiling and functional genomics studies can provide insights into the gene expression patterns and functional roles of key genes involved in seed size regulation. Additionally, the recent advances in AI-based models have allowed to simultaneous integration of multiple omics datasets, such as genomics, phenomics, single-cell genomics, enviromics, transcriptomics, epigenomics, metabolomics, and metagenomics, to provide a more holistic understanding of the complex regulatory networks controlling complex traits such as seed size in soybean (Figure 3) [100]. However, such models are in their initial phase but the use of these AI-based models in future will greatly assist in exploring the in-depth molecular regulatory network mechanism for seed size in soybeans [98].

## 5. Conclusions

Seed size traits of soybeans have a complex inheritance and molecular regulatory mechanism. Although, significant efforts have been made to elucidate the genetic basis of seed size in soybeans, and more than 400 QTLs related to seed size and shape of soybeans have been documented in SoyBase. Most of these QTLs have not been validated and are not used in soybean improvement. Additionally, the least efforts have been made to identify the candidate genes regulating seed size. Only a few genes have been isolated and their molecular mechanism has been unravelled for the seed size. However, the seed size is regulated by the complex network of pathways involving multiple genes, transcription factors, hormones, photoperiod, secondary metabolites, and other pathways. There is a dearth of systematic analysis of the molecular pathways upstream and downstream of seed size regulation in soybeans. Hence, considerable efforts are needed to unravel the regulatory network of pathways involved in the seed size regulation in soybeans. In this regard, the recent advances in AI-based models have allowed the use of multiomics datasets simultaneously, which in turn will assist in increasing the efficiency of the predictive breeding as well as exploration of the regulatory molecular mechanism of seed size in soybean. 

## Figures and Tables

**Figure 1 ijms-25-01441-f001:**
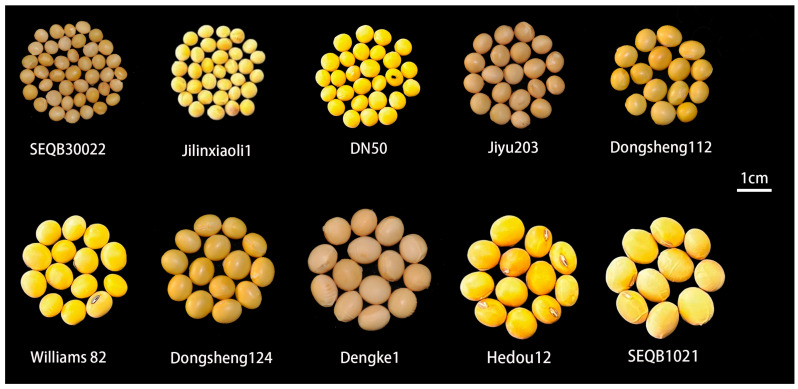
Seed size variation across the soybean germplasm. Soybeans exhibit abundant genetic diversity, often displaying significant variations in seed size and shape among different cultivars. This makes them an ideal material for investigating seed traits. Scale bar = 1 cm.

**Figure 2 ijms-25-01441-f002:**
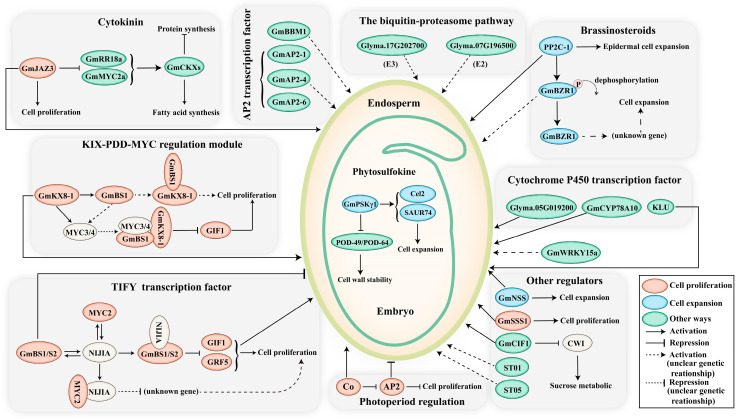
The major signalling pathways involved in soybean seed size control. The size of seeds is regulated through a complex network of signals, involving genes, hormones, transcription factors, and other intricate mechanisms. Dashed lines represent uncertain genetic relationships. Modulators that regulate seed size by influencing cell proliferation, cell expansion, and other regulatory processes are depicted in red, blue, and green, respectively. Abbreviations: BS, big seed; NIJIA, novel interactors of JAZ; NSS, novel seed size; SSS, soybean seed size; CWI, cell wall invertase; ST, seed thickness.

**Figure 3 ijms-25-01441-f003:**
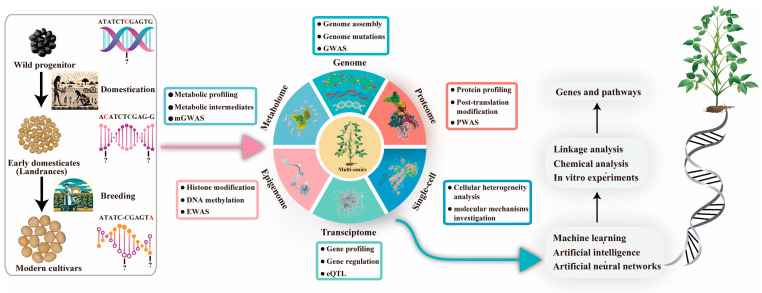
Recent high-throughput advances in genomics and other omics approaches have enhanced our understanding of genetic diversity. Integrating this data through AI and machine learning offers insights into the regulatory networks of key agricultural traits. Abbreviations: eQTL, expression qualitative trait locus; GWAS, genome-wide association study; EWAS, epigenome-wide association study; mGWAS, metabolite genome-wide association study; PWAS, proteome-wide analysis of SNPs; Single-cell, single-cell sequencing technology.

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
