# Peer review of "Understanding the Molecular Regulatory Networks of Seed Size in Soybean"

_ijms, 2024, doi:10.3390/ijms25031441_

Round 1
Reviewer 1 Report
Comments and Suggestions for Authors
Seed size of legumes is an important feature that influence the yield. In order to increase yield it is worth of studying the factors that influence the size of seeds. This review tells about the progress in research of molecular mechanisms that underlie in determination of soybean seed size. Authors give much information about genes involved in establishing a certain seed size; discuss the problems in study of molecular mechanisms of forming seed size and give the direction for future research.
I have some small but important comments:
(Interestingly beginning with the sheet 7, there is a new numeration of lines.)
Comments for the first part (before table 1)
Line 60 – “Rice has made substantial progress in this field” – this sounds as if rice, itself, has done progress. I guess it should be “research or rice”.
Lines 113-114 – give some references proving influence of environmental factors on soybean seed development.
Second part:
Line 143 – “been observed to results increased outer integument cell size in large seeds” – this sentence is not clear.
Check the text for A. thaliana. Somewhere Arabidopsis is used, somewhere - Arabidopsis thaliana. It should be A. thaliana everywhere after giving the full name. The same is with Medicago. Give the full name of Medicago truncatula and the use the short one – M. truncatula.
Author Response
We authors are thankful to the reviewer 1 and editor for their valuable suggestion and comments. These suggestions were very valuable for the improvement of our manuscript. The point-by-point response of the reviewer 1 comments are presented in the PDF file attached below, and the corrections made in the revised manuscript are highlighted by track changes and yellow color.

Reviewer 2 Report
Comments and Suggestions for Authors
The review paper is very interesting and contains valuable data for scientists working with plant model - soybean and symbiosis of this legume plant with rhizobia. The topic of the review is actual and interesting for many readers, since Soybean is a major cash crops which provides half of the vegetable oil and 25% of the plant proteins for human and animal food production. The authors have summarized comprehensive data regarding the molecular understanding of soybean seed size, which can be used in the future. The soybean seed size is an important trait affecting soybean yield.
In general, the work is of high scientific quality. It has been well written and presented. The authors have collected large data regarding this topic and cited large set of papers published within last decade. The work structure is good organized and the information included have been well and clearly described. The statements and conclusions are adequate to presented data and provided citations. The figures and table included into manuscript are appropriate, however their size could be enlarged, to be more easily for reading. However, a lack of information about the fact, that soybean plants establish symbiosis with Bradyrhizobia and this biological process essentially increases the production efficiency of soybean, is serious and these information should be added into the beginning of the Introduction section.
In conclusion, the work is of high scientific quality and should be interesting for readers of IJMS journal. I have only a few minor comments:
1) short information about symbiosis of soybean plants with rhizobia can be added into Introduction chapter together with appropriate citations;
2) the size of letters in table 1 should be enlarged;
3) the size of figures 1 and 2 should be increased.
Author Response
We authors are thankful to the reviewer 2 and editor for their valuable suggestion and comments. These suggestions were very valuable for the improvement of our manuscript. The point-by-point response of the reviewer 2 comments are presented in the PDF file attached below, and the corrections made in the revised manuscript are highlighted by track changes and yellow color.
